# *Annatto seeds* as Antioxidants Source with Linseed Oil for Dairy Cows

**DOI:** 10.3390/ani11051465

**Published:** 2021-05-20

**Authors:** Jesus A. C. Osorio, João L. P. Daniel, Jakeline F. Cabral, Kleves V. Almeida, Karoline L. Guimarães, Micheli R. Sippert, Jean C. S. Lourenço, Francilaine E. De Marchi, João P. Velho, Geraldo T. Santos

**Affiliations:** 1Department of Animal Science, State University of Maringa, Maringa, PR 87020-900, Brazil; jaco.mvz@hotmail.es (J.A.C.O.); jlpdaniel@uem.br (J.L.P.D.); kell-f@hotmail.com (J.F.C.); kleves_almeida@hotmail.com (K.V.A.); karolineg27@gmail.com (K.L.G.); micheli.sippert@gmail.com (M.R.S.); jeancarloslsss@gmail.com (J.C.S.L.); francieloise@hotmail.com (F.E.D.M.); 2Department of Animal Science, Santa Maria Federal University, Palmeira das Missões, RS 98300-000, Brazil; velhojp@ufsm.br

**Keywords:** polyunsaturated fatty acid, digestibility, lipoperoxidation, n-3

## Abstract

**Simple Summary:**

Currently, functional foods are gaining widespread attention. Polyunsaturated fatty acids (PUFA) and antioxidant compounds have beneficial effects on health. It is possible to increase the concentration of these compounds in the milk obtained from dairy cows by manipulating their diets, thereby improving milk quality and consequently the health of animals and humans who consume this milk. Annatto seed (*Bixa orellana* L.) is a source of antioxidants, whereas linseed oil is rich in omega 3 fatty acid. We evaluated the inclusion of annatto seeds and linseed oil in the diets of dairy cows and their effects on dry matter intake (DMI), nutrient digestibility, milk yield, milk composition and antioxidant capacity in milk and blood. There was no effect of treatment on nutrient digestibility and antioxidant capacity, but the addition of annatto seeds decreased DMI and milk production and linseed oil supplementation reduced milk fat content.

**Abstract:**

This study aimed to evaluate the effects of annatto seeds, linseed oil and their combination on DMI, apparent total tract digestibility, antioxidant capacity and milk composition of dairy cows. Four lactating Holstein cows (120 ± 43 days in milk; 15.98 ± 2.02 kg of milk/day, mean ± SD) were allocated in a 4 × 4 Latin square with a 2 × 2 factorial arrangement (with or without annatto seeds at 15 g/kg of dry matter (DM); with or without linseed oil at 30 g/kg of DM) and provided four different diets: control (no annatto seeds or linseed oil); annatto seeds (15 g/kg of DM); linseed oil (30 g/kg of DM); and a combination of both annatto seeds and linseed oil. Annatto seeds reduced DM intake, and milk yield, protein and lactose, but increased content of fat, total solids and short chain fatty acid, with no effect on total antioxidant capacity of milk. Linseed oil supplementation decreased medium chain fatty acid proportion and n-6/n-3 ratio, conversely it increased long chain fatty acids and n-3 fatty acid content of milk, ether extract intake and total-tract digestibility. Thus, linseed oil supplementation in dairy cow diets improved the milk FA profile but decreased milk fat concentration, whereas annatto seeds did not influence antioxidant capacity and depressed feed intake and milk yield.

## 1. Introduction

Foods containing a high concentration of n-3 fatty acids in the human diet can reduce the risk of cardiovascular disease and prostate, colon and breast cancer [1,2]. Hence, interest in the consumption of polyunsaturated fatty acids (PUFA)-rich dairy products has increased [3,4]. Cows receiving typical diets produce milk that contains approximately 70% saturated fatty acids (SFA), whereas mono- and polyunsaturated fatty acids represent approximately 25% and 5% of milk fat, respectively [5].

Linseed oil, which contains approximately 70% PUFA, of which 50% is α-linolenic acid (ALA), is a rich vegetable source of fatty acid (FA) n-3, and supplementation with linseed oil in cow diets leads to an increase in n-3 FA in their milk. [6]. The increase in the PUFA content of milk fat is desirable for human health; however, this makes the milk more susceptible to oxidation [7]. Because PUFA has double bonds, it is susceptible to the loss of electrons because of the action of free radicals, luminosity and other agents, leading to lipid peroxidation [8]. Lipid peroxidation promotes a rancid flavor and reduces the shelf life of dairy products [8] and can predispose humans to metabolic diseases [9,10]. In the animal body, there is a balance between the formation of free radicals and endogenous antioxidant capacity. Endogenous antioxidant capacity is regulated by the enzymes catalase (CAT), superoxide dismutase (SOD) and glutathione peroxidase (GPX), which can delay lipid peroxidation [11]. The body can also receive antioxidant compounds, such as vitamin E, selenium, phenolic compounds, from diet (exogenous antioxidants) [12,13].

The supply of these sources of antioxidants in dairy cow diets increases the antioxidant activity in milk [13,14]. The addition of carotenoids and vitamin E in the feeding of ruminants, can improve the organoleptic quality of the final product [15], both in meat and milk. This protects the product from the lipoperoxidation effects by improving the oxidative status of meat and milk [16]. Annatto (*Bixa orellana* L.) is a Bixaceae family plant, and its seeds contain carotenoids with antioxidants properties (Bixin and Norbixin). Bixin is a carotenoid that belongs to the apocarotenoid family [17], and whose antioxidant power is conferred by an extensive chain of double bonds, which allows it to combat singlet oxygens [18]. Annatto seeds have tocotrienols that, combined with bixin, could synergistically protect PUFAs from oxidation [19]. The objectives of this study were to evaluate the effects of dietary of annatto seeds, linseed oil, and their combination on dry matter intake (DMI) and nutrient total tract digestibility, as well as changes in the antioxidant capacity and milk FA composition of lactating cows. We hypothesized that linseed oil supplementation would increase n-3 concentration, whereas annatto seeds would increase antioxidant capacity and, consequently, decrease milk oxidation, and the association between the two treatments would increase n-3 concentration in the milk in conjunction with lower oxidation. 

## 2. Materials and Methods 

### 2.1. Cows, Diets, and Experimental Procedures

The experiment was conducted at the State University of Maringa (UEM), Brazil. The experimental protocol was approved by the UEM Ethics Committee (number 6450240117). Four multiparous Holstein cows (120 ± 43 days in milk, 15.98 ± 2.02 kg of milk/day, 566 ± 64 kg of body weight (BW), mean ± SD) were housed in a 4 × 4 Latin square design with a 2 × 2 factorial arrangement (with or without annatto seeds; with or without linseed oil), for a total of 84 days with four experimental periods of 21 days, with 16 days for adaptation and 5 days for collection). Diets were formulated for dairy cows averaging 580 kg of BW, 25 kg/day of milk with 35 g/kg of fat [20]. Treatments were formulated as follows (Table 1): (1) control (no added annatto seeds and no linseed oil; (2) annatto seeds (annatto seeds at 15 g/kg of dry matter (DM) added); (3) linseed oil (linseed oil at 30 g/kg of DM added); and (4) a combination of both annatto seeds and linseed oil added. The forage to concentrate ratio of the experimental diets was 600:400, on a DM basis. 

To ensure the intake of annatto seeds and linseed oil, before providing the total mixed ration (TMR)*,* these ingredients were weighed and mixed in the concentrate daily. The annatto seeds used contained 904 g/kg DM, 113 g/kg crude protein (CP), 51 g/kg ether extract (EE), 254 g/kg neutral detergent fiber (NDF) and have a total polyphenol content of 8.35 mg expressed as gallic acid equivalents (GAE)/g, total antioxidant capacity of 358 mM Trolox/mL, reducing power of 1.25 mg GAE/g and total antioxidant activity by capturing free radicals (DPPH) of 32.6 mg/mL. Animals were housed in a tie-stall and milking was performed twice daily. TMRs *(*Table 2) were offered after milking, with 60% of the total DM at 8:00 and the remaining (40%) at 16:00. Animals were weighed at the beginning and at the end of each period before the morning feed. The amount of feed supplied was adjusted to obtain 10% leftovers.

### 2.2. Dry Matter and Nutrients Intake and Digestibility

Intake was measured daily by weighing the feed provided and the refusals; feed intake was calculated for statistical analysis during the collection period (from day 17 to day 21) in which daily samples were collected and stored for further analysis. To estimate the total apparent digestibility of DM and food, fecal samples (100 g) were collected directly from the rectum in the following schedule: 17th day at 08:00 and 17:00; 18th day at 02:00, 11:00, and 20:00; 19th day at 05:00, 14:00, and 23:00, as described by Morris et al. [21]. Fecal samples were dried at 55 °C for 72 h and ground to pass through a 2-mm sieve to determine indigestible neutral detergent fiber (iNDF) and then through a 1-mm sieve for bromatological analysis.

Feed, refusals and fecal samples were pooled for each cow to obtain a composite sample per treatment and period. Samples of feed, refusals and feces were analyzed according to AOAC [22] for DM (DM, method 934.01), EE (method 920.85), ash (method 938.08), CP (method 981.10) and NDF according to Van Soest et al. [23]. The organic matter (OM) was calculated by the difference between the ash content and total DM. Non-fibrous carbohydrates (NFC) were obtained using the equation described by Sniffen et al. [24]: NFC = 100 − (CP + NDF + EE + MM), wherein MM is the mineral material. Indigestible NDF (iNDF) was used as an internal indicator to estimate daily fecal excretion. The iNDF was determined in the feed samples, refusals and feces, through its in situ incubation for 288 h into the rumen as described by Huhtanen et al. [25]. 

### 2.3. Lipid Profile and Blood Total Antioxidant Capacity

Blood samples were collected from the coccygeal vein on the 19th day of each period in tubes containing anticoagulant (BD Vacutainer^®^, K2EDTA 7.2 mg, São Paulo, SP, Brazil) 4 h after the morning feeding. After the sampling, samples were subjected to centrifugation at 1080× *g* for 15 min to obtain the plasma. Plasma was transferred to 2-mL vials and immediately frozen for later determination of triglycerides, total cholesterol, high density lipoprotein cholesterol (HDL-cholesterol), and blood total antioxidant capacity.

The blood concentrations of triglycerides, total cholesterol and HDL-cholesterol were determined using commercial kits (Gold Analisa Diagnóstica, Belo Horizonte, MG, Brazil), and the analyses were performed on a spectrophotometer (Bio-2000IL; Bioplus^®^, São Paulo, SP, Brazil). The determination of blood total antioxidant capacity was conducted according to Erel [26]. The samples (5 μL) were incubated with 200 μL of 0.4 M acetate buffer at pH 5.8. Subsequently, 20 μL of ABTS+-(2,2-azinobis-(3-ethyl-benzothiazolin-6-sulfonic acid)) ^•+^ solution was added in 30 mM acetate buffer at pH 3.6. The samples were incubated and absorbance was measured at 660 nm with a UV–vis spectrophotometer (Spectrum SP2000, Thermo Fisher, Waltham, MA, USA). Blood total antioxidant capacity was expressed in Trolox equivalent (μM Trolox/mL).

### 2.4. Milk Yield and Composition

Milk production was recorded with the use of meters coupled to milking equipment. For statistical analysis, only the data referring to the collection periods were used. On days 20 and 21, milk samples were collected and proportionally composited based on milk yield during the morning and afternoon milking. Then milk samples were divided into two aliquots. The first aliquot, approximately 50 mL of the milk, was kept at room temperature and stored with 2-bromo-2-nitropropane-1,3-diol (Bronopol, San Ramon, CA, USA) for the determination of fat, protein, lactose, defatted dry extract and milk density. The second aliquot, approximately 100 mL of milk, without the addition of preservative, was frozen at −80 °C for the subsequent determination of reducing power, total antioxidant activity total, conjugated diene hydroperoxides (CD), thiobarbituric acid reactive substances (TBARS), and composition of fatty acids.

The concentrations of nitrogen-urea, fat, protein, and lactose in milk were determined using a spectrophotometer (Bentley 2000; Bentley Instrument, Inc., Chaska, MN, USA) in the laboratory of the Dairy Analysis Program of the “Associação Paranaense dos Criadores de Bovinos da Raça Holandesa”, Curitiba, Pr, Brazil. The somatic cell count was obtained using an electronic counter (Somacount 500; Bentley Instrument, Inc., Chaska, MN, USA). Additionally, fat-corrected milk (4% FCM) was calculated using the Gaines [27] equation as follows: FCM = 0.4 × milk yield + 15 × fat yield.

### 2.5. Reducing Power of Milk

The reducing power was analyzed from the milk sample extracts obtained by adding 9 mL of methanol to 1 mL of milk. Thereafter, the mixture was vortexed for 5 min and centrifuged at 1080× *g* for 10 min. The total reducing power of milk was determined by a method described by Zhu et al. [28] with some modifications. Milk proteins were precipitated by adding 1 mL of a trichloroacetic acid solution (200:800; *v/v*) to 1 mL of milk. The mixture was vortex-mixed for 10 min and centrifuged at 1058× *g* for 10 min at 4 °C. Absorbance was measured at 700 nm on a UV–vis spectrophotometer (Spectrum SP2000, Thermo Fisher, Waltham, MA, USA), and reducing power was reported as GAE (mg/L).

### 2.6. Total Antioxidant Activity in Milk

Total antioxidant activity was analyzed from the milk sample extracts obtained by adding 9 mL of methanol to 1 mL of milk. Thereafter, the mixture was vortexed for 5 min and centrifuged at 1080× *g* for 10 min. Total antioxidant activity of the milk samples was determined by a method described by Rufino et al. [29], with the addition of radical ABTS+-(2,2-azinobis-(3-ethyl-benzothiazolin-6-sulfonic acid)) to the extract. Absorbance was measured at 734 nm on a UV–vis spectrophotometer (Spectrum SP2000, Thermo Fisher, Waltham, MA, USA) after 6 min of reaction. Total antioxidant activity was expressed in Trolox equivalent (μM Trolox/mL).

### 2.7. Conjugated Diene Hydriperoxides (CD) in Milk

The CD in milk were evaluated using the methodology described by Kiokias et al. [30], which reflects the lipid oxidation of milk. A 50 μL sample of milk was added to 2.5 mL of an isooctane/2-propanol solution (2:1, *v*/*v*) and vortexed for 1 min. Samples were filtered on a 0.22 mm PTFE membrane filter, and absorbance was measured at 232 nm on a UV–vis spectrophotometer (Spectrum SP2000, Thermo Fisher, Waltham, MA, USA). The CD were expressed in mmol/kg of fat.

### 2.8. Tiobarbituric Acid Reactive Substances (TBARS) in Milk

Analysis of TBARS was performed as described by Vyncke [31], with few adaptations. A 500 µL aliquot of milk was transferred into a 15-mL falcon tube (Cellstar, Greiner Bio-One, Americana, SP, Brazil) containing 2.0 mL of thiobarbituric acid solution (TBA 1%, TCA 15%, and 562.5 mM HCl). The samples were heated in a boiling water bath (100 °C) for 15 min, cooled in ice water for 5 min and then centrifuged at 1080× *g* for 10 min. The supernatant was transferred to a cuvette for later reading at 532nm on a UV–vis spectrophotometer (Spectrum SP2000, Thermo Fisher, Waltham, MA, USA). The results were expressed in concentration (mmol/kg fat).

### 2.9. Composition of Fatty Acids in Milk

To determine the milk fatty acid profile, fat was extracted by centrifugation, according to the methodology described by Murphy et al. [32] and fatty acids were esterified according to the ISO 5509 method [33] using KOH/methanol and n-heptane. Fatty acid methyl esters were quantified using gas chromatography (Trace GC 52 Ultra, Thermo Scientific, West Palm Beach, FL, USA) with self-sampling, equipped with a flame ionization detector at 240 °C and a fused silica capillary column (100 m in length, 0.25 mm internal diameter and 0.20 μm; Restek 2560, Thermo Scientific). The gas flow was 45 mL/min for H_2_ (carrier gas), 45 mL/min for N_2_ (auxiliary gas), and 45 a 400 m/min of synthetic air (flame gases). The initial temperature of the column was set at 50 °C, maintained for 4 min, remained from at 10 °C to then increased to 200 °C, and was maintained for 15 min. Then, it was maintained at 20 °C attaining reaching 240 °C and maintained for 8 min at the final temperature. The quantification of fatty acids in the sample was conducted by comparison with the retention time of fatty acid methyl esters from standard samples (18919-1 Sigma Aldrich, São Paulo, SP, Brazil).

### 2.10. Statistical Analysis

Statistical analysis was performed using the MIXED procedure of SAS version 9.3 (SAS Institute, Inc., Cary, NC, USA) according to the following model:*Y_ijkl_* = *μ* + *A_i_* + *O_j_* + *A × O_ij_* + *p_k_* + *a_l_* + *e_ijkl_*.
With *p_k_* ≈ *N* (0, $\sigma_{p}^{2}$), *a_l_* ≈ *N* (0, $\sigma_{a}^{2}$) e *e_ijkl_* ≈ *N* (0, $\sigma_{e}^{2}$), where *Y_ijkl_* is the observed value; *μ* is the general mean; *A_i_* is the fixed effect of annatto (*i =* 1 or 2); *O_j_* is the fixed effect of linseed oil (*j =* 1 or 2); *A × O_ij_* is the fixed effect of the interaction between annatto and linseed oil; *p_k_* is the random period effect; *a_l_* is the random effect of the animal; *e_ijkl_* is the residual error; *N* indicates the normal distribution; and $\sigma_{p}^{2}$, $\sigma_{a}^{2},$ and $\sigma_{e}^{2}$ are the variances associated with the random effects associated with the period and animal, and residual variance, respectively. Fisher’s least significant difference test (LSD) was applied when there was an interaction between the oil and annatto factors. For all analysis, the significance level was *p* ≤ 0.05, and trends were declared at 0.05 < *p* ≤ 0.10.

## 3. Results

There was no interaction between annatto seeds and linseed oil on DM and nutrient intake except on EE intake (*p* = 0.03) (Table 2). Indeed, DM, OM, CP, NDF and NFC intake was reduced by annatto seeds addition (*p* < 0.001), and linseed oil reduced the NFC intake by 8% (*p* = 0.03). There was no interaction between evaluated factors on DM and nutrients digestibility. In addition, annatto seeds did not affect digestibility. However, supplementation of linseed oil increased EE digestibility (*p* = 0.005), tended to decrease NDF digestibility (*p* = 0.07) and to increase CP digestibility (*p* = 0.09).

There was no interaction between annatto seeds and linseed oil on milk yield, composition and antioxidant activity (Table 3). Diets containing annatto seeds reduced the production of milk (*p* = 0.01), protein (*p* = 0.01), and lactose (*p* = 0.01). Annatto seed addition increased the fat content and total solids content (*p* < 0.01) and supplementation with linseed oil decreased milk fat content (*p* = 0.03).

An interaction effect between annatto seeds and linseed oil was observed in milk C6:0 (*p* = 0.007), C8:0 (*p* = 0.01), and C18:2 n6t (*p* = 0.04) (Table 4). The addition of annatto seeds increased the concentrations of C6:0 and C8:0 fatty acids in the absence of linseed oil (*p* ≤ 0.05), while in the presence of oil the concentrations of these fatty acids was not changed. Supplementation of linseed oil increased the concentrations of C18:2 n6t FA (*p* ≤ 0.05), while in combination with annatto seeds, the concentration of this FA was not affected. It was observed that linseed oil supplementation reduced the concentration of C10:0, C12:0, C13:0, C14:1, C15:0, and C16:0 in milk fat (*p* < 0.05), and increased the C18:0, C18:1 n9t, C18:2 n6t, C18:3 n3, C18:2 c9 t11 -CLA, C20:3 n6, C20:4 n6, and C21:0 (*p* < 0.05).

An interaction effect between annatto seeds and oil was observed on short chain fatty acids (SCFA) (*p* < 0.01), addition of annatto seeds alone increased milk SCFA in the absence of linseed oil (*p* ≤ 0.05), while in the presence of oil it did not change the concentrations of these fatty acids (Table 5). Linseed oil decreased medium chain fatty acids (MCFA) (*p* = 0.01) and increased long chain fatty acids (LCFA) (*p* = 0.02) in milk fat. Additionally, linseed oil supplementation tended to reduce mono-unsaturated fatty acids (MUFA) (*p* = 0.07) and increase PUFA (*P*= 0.08) and saturated fatty acids (SFA) (*p* = 0.06). There was a tendency for interaction between annatto seeds and linseed oil on *fatty acids* n-6 (*p* < 0.10), wherein linseed oil supplementation increased the *fatty acids* n-6 in the absence of annatto but in the presence of annatto seeds this fatty acid was reduced (Table 5). There was also an approximately 176% increase of n-3 concentration in milk fat (*p* < 0.01) and the n-6/n-3 ratio was improved by linseed oil supplementation once the linseed oil was reduced by 58% (*p* < 0.01).

The different treatments, as well as the addition of annatto seeds did not have any effect on the antioxidant concentrations in milk (Table 3) or blood (Table 6). The concentration of CD in milk has a tendency to increase with addition of linseed oil (*p* = 0.07) to the diet. There was no effect of annatto seeds on blood parameters, linseed oil supplementation has a tendency to increase HDL-cholesterol (*p* = 0.08).

## 4. Discussion

The addition of annatto seeds to the cow diet reduced feed intake. It was observed that cows rejected diets containing annatto seeds. It was associated with undesirable palatability, texture, and odor. Additionally, annatto seeds contain carotenoids, terpenoids, and terpenes [34,35,36], which affect the DMI [37,38]. Terpenoids and terpenes can be stored in the form of essential oils and released through structures, such as secretory glands and trichomes; consequently, these compounds are concentrated in the taste buds [39]. The main essential oil components of annatto seeds are two monoterpenes called α-pinene and β-pinene [40]. According to Estell et al. [41], α-pinene in the alfalfa pellets rendered them unpalatable for consumption by sheep. It is possible that the monoterpene α-pinene may be associated with animal rejection of annatto seeds.

The linseed oil supplementation, regardless of annatto seed supply, increased EE intake and reduced the NFC intake. According to the NRC [20], increased dietary energy density by fat supplementation leads to a replacement of carbohydrates by lipids, which might result in a higher EE content in the diet and a lower proportion of carbohydrates. Lipid supplementation increased the digestibility of EE and CP. Similar results were found by Santos et al. [42] when dairy cow diets were supplemented with 2.5% linseed oil. Linseed oil had a higher proportion of PUFA and a low melting point facilitating micelle formation and increasing absorption rate in the intestine [20]. Thus, the addition of linseed oil to diets increased the digestibility of EE.

Exceeding 5% of the dietary lipid DM may cause some adverse effects, such as inhibiting the growth of microorganisms that degrade the fibrous fraction in the roughage and affecting fiber and OM digestibility [43]. The percentage of EE in diets with oil was 5.6%, which may explain the tendency to reduce NDF digestibility. However, exceeding the 7% lipid limit can have more evident effects on reducing DMI, and digestibility of DM, OM, and NDF [44].

In the present study, 15 g/kg of annatto seeds in DM did not alter nutrient digestibility. These results were consistent with the findings of Barbosa et al. [45] who used higher amounts of annatto seeds (100 g/kg, 230 g/kg, and 350 g/kg of DM) in diets for sheep and found that annatto seeds did not influence nutrient digestibility. De Lima Júnior et al. [46] reported increasing levels (0, 100, 200, and 300 g/kg total DM) of the annatto by-product in the sheep feed did not affect the digestibility of the layers.

In this study, the treatments did not affect the lipid profile (cholesterol, triglycerides, and HDL-cholesterol). These parameters followed the standard values according to Kaneko et al. [47]. Although some studies have highlighted the potential effects of antioxidants on these parameters, this was not observed in the present study [48,49]. Similar results were reported by Lima et al. [50], where supplementation of the diet with annatto colorum (paprika) in 0.08, 0.12, and 0.16 g/kg of DM for lactating cows did not affect plasma cholesterol concentration but altered the plasma fatty acid profile. In the present study, linseed oil supplementation increased the HDL-cholesterol, which could be attributed to lipid intake.

Milk and lactose production were reduced with diets containing annatto seeds, which also decreased DM and nutrient intake. According to Dado and Allen [51], milk yield is positively correlated with DMI, resulting in lower net energy intake available for milk synthesis in comparison with the control diet, which had higher production because of higher DMI. Additionally, the lower DMI decreased glucose precursors. As synthesis of lactose is dependent on glucose, lactose is comprised in annatto seed-fed animals [52]. Milk fat and total solid content increased after consumption of annatto seeds diets. These effects occurred probably because of to a concentration effect because these diets showed a 296 reduction in milk yield [53].

The addition of annatto seeds did not affect the proportions and ratios of fatty acids in milk, especially the MCFA that are responsible for increasing the fat content, suggesting that these amounts of annatto seeds were not the main cause of the increased content of fat. Additionally, fat-corrected milk production and milk fat production did not show significant effects related to annatto seeds and lipid supplementation.

The addition of annatto seeds did not affect the oxidation products and antioxidant activity, as the seeds were included with the objective of transferring its polyphenolic compounds to milk, to decrease oxidation of milk fat that was enriched with PUFA. Treatments with annatto seeds had approximately 125.25 mg GAE/g DM of total polyphenols, which was not satisfactory to affect concentrations, increase antioxidant activity, or decrease the oxidative profile. Cows supplemented with 24.35 mg GAE/kg DM of total polyphenols from propolis extract did not improve the oxidative stability of milk [20]. Conversely, Santos et al. [54] found no significant difference for the oxidative profile but found increased antioxidant activity with the supplementation of 1950 mg GAE/g DM of total polyphenols from grape residue in the presence of soybean oil.

Regarding the antioxidant activity in blood and milk, lipid supplementation increased CD concentration because of increased PUFA intake, which has double bonds that predispose it to lipoperoxidation, thus causing electron loss [55] and increased CD levels in milk. Supplementation with annatto seeds did not affect the animal or milk oxidative status, which may have been caused by the low absorption range of carotenoids. Bixin is a carotenoid that belongs to the group of carotenes, whose absorptive process is very similar to that of lipids, because carotenes are fat-soluble pigments [56]. Carotenes are less polar molecules found in the center of emulsions, whose absorptive efficiency depends on the transfer of the emulsion to the micelles, which is not as effective for the carotene group compared to the xanthophyll group [56].

Supplementation with linseed oil reduced the fat content in milk. The PUFA content in the linseed oil is responsible for this reduction [57,58] by altering ruminal biohydrogenation resulting in intermediate products, such as fatty acids trans-9, cis-11-CLA, and trans-10, cis-12-CLA, which also inhibit the synthesis of milk fat in the mammary glands [59,60]. However, consumption of material and EE was lower in diets supplemented with annatto seeds, according to Woolpert et al. [61], and diets with lower EE content led to an increase in de novo fatty acid synthesis. This is the main source of the formation of SCFA in the mammary glands.

Linseed oil diet supplementation reduced the MCFA, owing to the higher concentration of PUFAs in flax oil [62]. Linseed oil PUFAs can be affected by incomplete biohydrogenation generating the trans-9, cis-11-CLA, and trans-10, cis-12-CLA [59,60]. These fatty acids are inhibitors of genes that are involved in the de novo synthesis [59,60], such that these CLAs reduced MCFA (C10:0, C12:0, C13:0, C14:1, C15:0, C16:0) and led to a tendency to reduce C14:0. It is noteworthy that these fatty acids are of mixed origin, originating from the de novo synthesis that occurs in the mammary gland and from food.

In general, supplementation with linseed oil increased the LCFA (C18:0, C18:1 n9t, C18:2 n6t, C18:3 n3, C20:3 n6, C20:4 n6, and C21:0). This can be explained by the composition of linseed oil, which is a source of LCFA and PUFA, mainly C18:3 n-3. When the PUFA reaches the rumen, it can be affected by biohydrogenation to protect ruminal bacteria generating intermediates and saturated fatty acids as a final product. The fatty acids that leave the rumen are absorbed in the intestine and incorporated into milk fat [63]. The C18:2 n6t is produced during the biohydrogenation process, and its reduction may be associated with the addition of carotenoids. Hino et al. [64] performed an in vitro assay, in which they added 5 or 10 mg of β-carotene, 5 mg α-tocopherol, and 1 g of glucose for each liter of rumen liquid, and observed the growth of ruminal microorganisms. The association of β-carotene and α-tocopherol with sunflower oil decreased the inhibition of the growth of these microorganisms caused by oil; thus, increasing the use of LFCA by ruminal bacteria. Possibly the annatto seed carotenoids promoted the use of the C18:2 n6t fatty acid in the presence of oil, resulting in a lower concentration of n-6 compared to the diet with flaxseed oil.

Supplementation with linseed oil decreased SFA and increased MUFA and PUFA. The increase in PUFA results in a better fatty acid profile for human consumption. The results of the present study are consistent with the findings of Santos et al. [42] and Suksombat et al. [65], who reported similar changes in milk fatty acids when linseed oil was used, and those of Caroprese et al. [66] and Petit and Côrtes [67], who used linseed to feed dairy cows.

Linseed oil is a rich source of C18:3 n-3, which is partially hydrogenated, and a large part of it becomes overpassed, which promotes an increase in the concentration of n-3 in milk fat. There was an increase of approximately 176% in the concentration of n-3 in milk obtained from cows fed linseed oil supplemented diets as compared to those fed normal diets. Previous studies demonstrated an increase of n-3 concentration in milk from dairy cows receiving linseed oil supplementation [42,58,68]. Fatty acids n-3 has a beneficial effect on human health owing to its potential to reduce cardiovascular disease [1], prostate, colon, and breast cancer [2]. Oil supplementation in the cow diet improves the n6:n3 ratio by 58%, which makes their milk healthier, according to the World Health Organization [69].

## 5. Conclusions

The addition of 15 g/kg DM of annatto seeds did not affect milk antioxidant capacity but reduced feed intake and milk yield. The addition of 30 g/kg DM of linseed oil decreased milk fat content and the ratio between fatty acids n-6:n-3.

## Figures and Tables

**Table 1 animals-11-01465-t001:** Ingredients and chemical composition of total mixed diets.

Variables	Diets			
	No Linseed Oil		With Linseed Oil	
	No Annatto	With Annatto	No Annatto	With Annatto
Ingredients				
Corn silage (g/kg of DM ^1^)	600	600	600	600
Ground corn (g/kg of DM)	202.8	189.1	166.1	152.5
Soybean meal (g/kg of DM)	165.2	163.9	171.9	170.5
Linseed oil (g/kg of DM)	0	0	30	30
Mineral and vitamin ^2^ (g/kg of DM)	22	22	22	22
Annatto Seeds (g/kg of DM)	0	15	0	15
Molasses Powder (g/kg of DM)	5	5	5	5
Limestone (g/kg DM)	5	5	5	5
Chemical composition				
Dry matter (g/kg fresh weight)	453	453	448	448
Organic matter (g/kg of DM)	934	933	934	933
Crude protein (g/kg of DM)	145	145	146	146
Ether extract (g/kg of DM)	27.8	28.0	56.4	56.5
Neutral detergent fiber (g/kg of DM)	332	334	329	331
Non-fibrous carbohydrates (g/kg of DM)	429	428	401	400
NE_L3x_ ^3^ (MJ/kg of DM)	6.61	6.61	6.92	6.92

^1^ DM—Dry matter; ^2^ Composition of mineral and vitamin supplement (per kg of product)—145 g of calcium, 51 g of phosphorus, 20 g of sulfur, 33 g of magnesium, 28 g of potassium, 93 g of sodium, 30 mg of cobalt, 400 mg of copper, 10 mg of chromium, 2000 mg of iron, 40 mg of iodine, 1350 mg of manganese, 15 mg of selenium, 1700 mg of zinc, 510 mg of fluorine; ^3^ NE_L3x_ (MJ/kg of DM) = [0.0245 × TDN_1x_ (g/kg) − 0.12] × 4.184, were total digestible nutrients(TDN) were estimated as: TDN_1X_ (g/kg) = g/kg digestible NFC + g/kg digestible CP + g/kg digestible NDF + (g/kg digestible EE × 2.25).

**Table 2 animals-11-01465-t002:** Dry matter and nutrients intake and digestibility of Holstein cows fed diets containing annatto seeds and/or linseed oil.

Variables	Diets				SEM ^7^	*p*-Value		
	No Linseed Oil		With Linseed Oil			Anna ^8^	Oil ^9^	Int ^10^
	No Annatto	With Annatto	No Annatto	With Annatto				
Intake (kg DM/day)								
DM ^1^	14.12	12.30	13.76	12.20	0.192	<0.001	0.13	0.30
OM ^2^	13.25	11.47	12.87	11.39	0.181	<0.001	0.17	0.34
CP ^3^	2.11	1.84	2.04	1.86	0.023	<0.001	0.36	0.24
EE ^4^	0.40 ^c^	0.35 ^d^	0.76 ^a^	0.69 ^b^	0.015	<0.001	<0.001	0.03
NDF ^5^	4.53	3.89	4.38	3.87	0.091	<0.001	0.34	0.47
NFC ^6^	6.22	5.39	5.67	5.01	0.101	<0.001	0.004	0.33
Total Apparent Digestibility (g/kg of DM)								
DM ^1^	0.69	0.69	0.68	0.68	0.014	0.86	0.14	0.67
OM ^2^	0.71	0.71	0.71	0.72	0.017	0.77	0.96	0.74
CP ^3^	0.72	0.72	0.74	0.74	0.019	0.76	0.09	0.85
EE ^4^	0.82	0.82	0.87	0.86	0.012	0.73	0.005	0.62
NDF ^5^	0.56	0.55	0.53	0.55	0.013	0.74	0.07	0.21
NFC ^6^	0.81	0.82	0.82	0.82	0.014	0.83	0.85	0.92

^1^ DM—Dry matter; ^2^ OM—Organic matter; ^3^ CP—Crude protein; ^4^ EE—Ether extract; ^5^ NDF—Neutral detergent fiber; ^6^ NFC—Non-fiber carbohydrate; ^7^ SEM—Standard error of mean; ^8^ Anna—Annatto seed; ^9^ Oil—Linseed oil; ^10^ Int—Interaction annatto seed and Linseed oil; ^a–d^ Fisher means test (LSD) at 5% of probability.

**Table 3 animals-11-01465-t003:** Production, composition, oxidative profile and antioxidant activity of milk from Holstein cows feed annatto seeds and/or linseed oil.

Variables	Diets				SEM ^9^	*p*-Value		
	No Linseed Oil		With Linseed Oil			Anna ^10^	Oil ^11^	Int ^12^
	No Annatto	With Annatto	No Annatto	With Annatto				
Production								
Milk yield (kg/d)	18.20	14.42	17.15	14.15	1.452	0.01	0.49	0.67
FCM (kg/d) ^1^	17.28	15.18	14.64	14.93	1.182	0.39	0.19	0.27
Fat (kg/d)	0.58	0.55	0.44	0.54	0.031	0.42	0.12	0.16
Protein (kg/d)	0.58	0.48	0.60	0.47	0.044	0.01	0.90	0.73
Lactose (kg/d)	0.81	0.63	0.76	0.61	0.065	0.01	0.37	0.66
Composition (g/kg of DM)								
Fat (g/kg)	33.87	39.65	26.85	38.62	1.152	<0.001	0.03	0.10
Protein (g/kg)	32.62	34.07	35.32	33.60	0.713	0.92	0.43	0.27
Lactose (g/kg)	45.02	43.70	44.25	43.57	0.034	0.08	0.39	0.53
TS (g/kg) ^2^	110.5	117.4	106.42	115.8	1.745	0.01	0.25	0.60
MUN (mg/dl) ^3^	12.83	11.22	13.63	13.11	0.918	0.25	0.17	0.55
SCS ^4^	2.14	2.39	2.29	2.30	0.091	0.42	0.84	0.47
Oxidative products and antioxidant activity								
CD ^5^	47.47	45.79	59.50	56.68	2.621	0.68	0.07	0.91
TBARS ^6^	13.23	14.72	16.71	13.67	1.002	0.57	0.38	0.12
TAC ^7^	241.0	242.8	239.9	252.8	2.253	0.78	0.73	0.86
Reducing power ^8^	34.37	34.71	33.32	36.96	2.673	0.67	0.89	0.72

^1^ FCM—Fat-corrected milk; ^2^ TS—Total solids; ^3^ MUN—Milk urea nitrogen; ^4^ SCS—Somatic cell score (log somatic cell count); ^5^ CD—Conjugated dienes (mmol/kg of fat); ^6^ TBARS—Thiobarbituric acid reactive substances (mmol of malondialdehyde/kg of fat); ^7^ TAC—Total antioxidant capacity (μM of Trolox equivalent/mL); ^8^ Reducing power—(mg of gallic acid equivalent/L); ^9^ SEM—Standard error of mean; ^10^ Anna—Annatto seed; ^11^ Oil—Linseed oil; ^12^ Int—Interaction annatto seed and linseed oil.

**Table 4 animals-11-01465-t004:** Composition of fatty acids in milk (g/100g of total lipids) of Holstein cows fed annatto seeds and linseed oil.

Variables	Diets				SEM ^2^	*p*-Value		
	No Linseed Oil		With Linseed Oil			Anna ^3^	Oil ^4^	Int ^5^
	No Annatto	With Annatto	No Annatto	With Annatto				
C6:0	0.17 ^b^	0.64 ^a^	0.33 ^b^	0.16 ^b^	0.021	0.02	0.01	0.007
C8:0	0.44 ^b^	0.75 ^a^	0.43 ^b^	0.29 ^b^	0.042	0.20	0.01	0.01
C10:0	2.25	2.64	1.65	1.49	0.182	0.76	0.04	0.43
C11:0	0.04	0.20	0.04	0.05	0.021	0.03	0.04	0.06
C12:0	3.64	3.58	2.73	2.36	0.221	0.58	0.03	0.67
C13:0	0.19	0.19	0.15	0.10	0.011	0.32	0.01	0.18
C14:0	13.66	12.09	10.85	10.25	0.564	0.35	0.08	0.69
C14:1	1.18	1.19	1.01	0.71	0.071	0.23	0.02	0.19
C15:0	1.18	1.07	0.99	0.86	0.052	0.18	0.04	0.90
C15:1	0.02	0.02	0.07	0.01	0.011	0.30	0.50	0.31
C16:0	33.12	30.68	26.21	27.35	0.871	0.69	0.02	0.33
C16:1	1.89	2.08	1.53	1.72	0.022	0.36	0.12	0.72
C17:0	0.75	0.78	0.71	0.71	0.021	0.71	0.40	0.71
C17:1	0.32	0.31	0.17	0.22	0.043	0.82	0.17	0.74
C18:0	10.53	10.85	13.25	15.81	0.562	0.25	0.01	0.36
C18:1 n9t	4.18	3.73	7.62	5.42	0.391	0.07	0.005	0.19
C18:1 n9c	23.96	25.69	27.14	28.52	1.224	0.42	0.16	0.77
C18:2 n6t	0.11 ^b^	0.09 ^b^	0.01 ^a^	0.20 ^b^	0.03	0.030	0.002	0.04
C18:2 n6c	1.52	1.55	1.73	1.40	0.065	0.23	0.82	0.18
C18:3 n6	0.06	0.05	0.05	0.04	0.010	0.09	0.25	0.90
C18:3 n3	0.10	0.10	0.34	0.23	0.010	0.18	0.002	0.19
C18:2 c9 t11-CLA ^1^	0.68	0.72	1.23	0.94	0.182	0.44	0.03	0.30
C18:2 t10 c12-CLA ^1^	0.04	0.03	0.06	0.04	0.021	0.40	0.40	0.53
C20:0	0.12	0.13	0.13	0.16	0.010	0.18	0.16	0.50
C20:1	0.04	0.05	0.07	0.05	0.011	0.74	0.25	0.30
C20:2	0.01	0.02	0.03	0.03	0.010	0.75	0.31	0.50
C20:3 n6	0.04	0.04	0.02	0.03	0.010	0.62	0.01	0.33
C20:3 n3	0.01	0.01	0.01	0.01	0.010	0.40	0.41	0.44
C20:4 n6	0.12	0.12	0.07	0.09	0.012	0.27	0.005	0.18
C20:5 n3	0.01	0.01	0.01	0.01	0.010	0.54	0.79	0.72
C21:0	0.60	0.55	0.96	0.73	0.091	0.21	0.03	0.42
C22	0.01	0.01	0.01	0.01	0.010	0.99	0.75	0.57
C24:0	0.02	0.02	0.02	0.01	0.010	0.89	0.52	0.16
C24:1	0.02	0.02	0.01	0.01	0.010	0.56	0.07	0.51

^1^ CLA—conjugated linoleic acid; ^2^ SEM—Standard error of mean; ^3^ Anna—Annatto seed; ^4^ Oil—Linseed oil; ^5^ Int—Interaction annatto seed and linseed oil; ^a,b^ Fisher means test (LSD) at 5% of probability.

**Table 5 animals-11-01465-t005:** Percentage concentrations and ratios of fatty acids grouped in the milk of Holstein cows fed annatto seeds and linseed oil.

Diets					SEM ^8^	*p-*Value		
-	No Linseed Oil		With Linseed Oil			Anna ^9^	Oil ^10^	Int ^11^
	No Annatto	With Annatto	No Annatto	With Annatto				
Total CLA ^1^	0.70	0.75	1.32	0.99	0.155	0.40	0.02	0.28
SCFA ^2^	0.61 ^b^	1.41 ^a^	0.76 ^b^	0.45 ^b^	0.063	0.05	0.01	0.002
MCFA ^3^	55.28	51.66	43.69	43.51	1.492	0.47	0.01	0.62
LCFA ^4^	44.11	46.94	55.55	56.38	1.663	0.58	0.02	0.79
SFA ^5^	69.37	67.07	60.11	62.08	1.391	0.89	0.07	0.47
MUFA ^6^	28.67	30.97	39.27	35.92	1.262	0.8	0.08	0.52
PUFA ^7^	1.97	1.96	2.63	2.00	0.083	0.09	0.06	0.10
n-3	0.12	0.11	0.36	0.25	0.011	0.20	0.001	0.19
n-6	1.85	1.85	2.27	1.75	0.061	0.08	0.23	0.10
n-6/n-3	15.41	15.42	6.31	7.00	0.541	0.88	<0.001	0.61

^1^ CLA—Conjugated linoleic acid; ^2^ SCFA—Short chain fatty acids; ^3^ MCFA—Medium chain fatty acids; ^4^ LCFA—Long chain fatty acids; ^5^ SFA—Saturated fatty acids; ^6^ MUFA—Monounsaturated fatty acids; ^7^ PUFA—Polyunsaturated fatty acids; ^8^ SEM—Standard error of mean; ^9^ Anna—Annatto seed; ^10^ Oil—linseed oil; ^11^ Int—Interaction annatto seed and linseed oil; ^a,b^ Fisher means test (LSD) at 5% of probability.

**Table 6 animals-11-01465-t006:** Parameters and blood antioxidant capacity of Holstein cows fed annatto seeds and linseed oil.

Variables	Diets				SEM ^3^	*p-*Value		
	No Linseed Oil		With Linseed Oil			Anna ^4^	Oil ^5^	Int ^6^
	No Annatto	With Annatto	No Annatto	With Annatto				
Cholesterol (mg/dL)	95.37	102.88	101.00	97.85	4.792	0.82	0.97	0.59
Triglycerides (mg/dL)	9.00	10.50	9.00	8.50	0.826	0.51	0.74	0.33
HDL-Cholesterol (mg/dL) ^1^	66.00	62.62	71.87	74.50	2.155	0.93	0.08	0.51
TAC ^2^	313.14	337.30	318.46	336.33	24.72	0.50	0.94	0.91

^1^ HDL-Cholesterol—High density lipoprotein; ^2^ TAC—Total antioxidant capacity (μM of Trolox equivalent/mL); ^3^ SEM—Standard error of mean; ^4^ Anna—Annatto seed; ^5^ Oil—linseed oil; ^6^ Int—Interaction annatto seed and linseed oil.

## Data Availability

Data is contained within the article. The data presented in this study are available in Annatto seeds as Antioxidants Source with Linseed Oil for Dairy Cows.

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
