# Peer review of "Annatto seeds as Antioxidants Source with Linseed Oil for Dairy Cows"

_animals, 2021, doi:10.3390/ani11051465_

Round 1

Reviewer 1 Report

Despite this reviewer appreciate the effort the authors made in replying to some of his comments, several points have not been addressed. Furthermore, the replies to the comments this reviewer pointed out in the previous review round were intended to be included in the final manuscript. In fact, comments were made to improve the quality of the final manuscript. Despite that, authors replied to them in the cover letter without modifying the final manuscript in several cases (i.e. the reference this reviewer asked for the timing of feces collection, the request this reviewer made to move the p values at the end of each sentence in the results section and several others).   

As another general comment, M&M section should be split out to include sampling times, sample collection and analysis performed for each parameter considered in this study in a separate section. Notice that subheading must be informative to allow the reader to keep what the authors are referring to (i.e. “chemical composition and digestibility”…of what?; “parameters and blood antioxidant capacity”…which parameters?; “reducing power and antioxidant activity”…of which parameter?), furthermore, numbering of the subheadings is wrong in the present format. 

This reviewer strongly recommends an extensive language revision, as the manuscript is hard to follow in the present form.

As no power analysis has been provided, the question on the statistical power of the model respecting to the variability of the parameters measured remained unsolved. 

Reviewer 2 Report

The submitted manuscript (animals-1106261) reports a study investigating the effects of annatto seed and linseed oil supplementation on cows dry matter intake and nutrient digestibility, milk yield and composition, milk fatty acids profile, and oxidative stability and activity in blood and milk. The results showed that linseed oil supplementation improved the milk fatty acid profile, increasing the n-3 concentrations and annatto seed addition did not influence the milk antioxidant capacity. Overall, layout of the manuscript is robust and the main assumptions are put forward on a scientifically solid basis. However, the following points should be addressed:

Line 40; EE should be defined at first mention.

Line 41; “improves” should be changed to “improved”.

Line 95, 97; BW, DM should be defined at first mention.

Line 109; What the “TMR” means? Please explain it.

Line 117-119; the author say that collected feces directly from the rectum in the following schedule: 17th day at 08:00 and 17:00, 18th day at 02:00, 11:00 and 20:00, 19th day at 05:00, 14:00 and 23:00. why choose these time to collect? Can you provide some evidences?

Line 126; HDL should be defined at first mention.

Line 202-204; What the “ raised from 10 °C to 10 °C to 200 °C, raised from 20 °C to 20 °C reaching 240 °C” means? It makes me feel puzzled. Please make it clear.

Line 210, 213; Is the “eijk” or “eijkl” ? Please be consistent.

Line 217; what the “ate” means? It should be changed to “at”.

Line 220; OM should be defined at first mention.

Line 245; why change the “p” to “P”? it should be changed back to “p”. and also the SFA should be defined at first mention.

Line 376-379; The conclusion should be in the past tense, not the present tense.

Reviewer 3 Report

I can understand that you added linseed oil on diets for dairy cows improves the milk fatty acid profile, increasing the n-3 concentrations and annatto seed addition did not influence the milk antioxidant capacity. The association of annatto seed and linseed oil did not influence the evaluated parameters.
However, the addition of annatto decreases DM intake and milk production. There was no effect of the treatments on digestibility of nutrients and linseeds oil supplementation reduced the milk fat content.
How do you think annatto seed and linseeds oil are useful in dairy farming industry ?
We should drink milk and eat annatto seed or linseeds oil directly for human health ?
How much should we drink CLA rich milk for preventing cancer ? I think that is worse for our health.

You need to have this manuscript proofread by a native speaker (researcher).

My specific comments are as follows,
L8; (J.A.C.O); ); --- (J.A.C.O);
L20; orellana L) --- orellana L.)
L22; dry matter --- dry matter (DM)
L27; dry matter intake --- DM intake
L38; EE --- first appearance ?
L65; orellana L) --- orellana L.)
L70; dry matter --- dry matter (DM)
L80; Holstein (120 --- Holstein cows (120
L88; 15 g/kg of DM) --- 15 g/kg of DM added)
30 g/kg of DM) --- 30 g/kg of DM added),
and linseed oil. --- and linseed oil added.
Table 1 --- Dry matter --- DM ?
L97; TMR --- first appearance ?
L111; (Vacutainer®) --- (model No., Company, Place)
K2EDTA --- What is this ?
L112; 3000 x g for 15 minutes --- L155; 1080 g x 10 minutes --- L158; 1058 g for 10 min --- L177; 3000 x g for 10 minutes --- Which is true ?
L114; HDL --- first appearance ?
L115; (Waikato MKV) --- (model No., Company, Place)
L120; (Bronopol)for --- (Bronopol) for
L132; MM). --- What is MM ?
L138; HDL --- HDL-cholesterol ? or High Density Lipoprotein ?
L139; (Gold Analisa®, Belo Horizonte) --- (model No., Company, Place)
L140; (Bioplus 2000®, São Paulo, SP) --- (model No., Company, Place)
L149; (Somacount 500®, Chaska, MN) --- (model No., Company, Place)
L159; (Spectrum SP2000) --- (model No., Company, Place)
L160; Total antioxidant capacity (TAC) --- See L114.
L162; (Spectrum SP2000) --- (model No., Company, Place)
L170; (Spritzen, Shanghai, China) --- (model No., Company, Place)
L171; (Spectrum SP2000) --- (model No., Company, Place)
L174; falcon tubes --- (model No., Company, Place) --- for example (Falcon®, Corning Inc..... ?)
L178; (Spectrum SP2000) --- (model No., Company, Place)
L178; in (mmol / kg fat) concentration. --- in concentration (mmol / kg fat).
L186; Thermo Scientific --- Thermo Fisher Scientific Inc. ?
L188; Restek 2560 --- (Company, Place) ...Restek Corporation ?
L194; (Sigma Aldrich).  --- (model No., Company, Place) --- Please show us in detail for we can order it if some researchers will give a supplementary exam.
L197; SAS --- Did not you refer SAS user's guide ?
L203; It has been theoretically confirmed that the LSD method is not suitable as a multiple comparison method. I think you should use Tukey, Dunnett, Duncan for multiple comparison especially among more than 3 treatments.
L220; increased (p ≤ 0.05) the concentrations of C6:0 and C8:0 acids in --- increased the concentrations of C6:0 and C8:0 acids (p ≤ 0.05) in

L228; An interaction effect (p < 0.01) between annatto seed and oil was observed on SCFA Annatto seed increased (p ≤ 0.05) milk SCFA in the absence of linseed oil, while in the presence of oil it did not (p > 0.05) change the concentrations of these acids fatty (Table 5). --- An interaction effect between annatto seed and oil was observed on SCFA (p < 0.01). Annatto seed increased milk SCFA (p ≤ 0.05) in the absence of linseed oil, while in the presence of oil it did not change the concentrations of these fatty acids (p > 0.05) (Table 5).

L230; Linseed oil decreased (p = 0.01) MCFA and increased (p = 0.02) LCFA in milk fat. --- Linseed oil decreased MCFA (p = 0.01) and increased LCFA (p = 0.02) in milk fat.

L231; In addition, linseed oil supplementation tended to reduce (p = 0.07) MUFA and increase (P= 0.08) PUFA and (p = 0.06) SFA. --- In addition, linseed oil supplementation tended to reduce MUFA (p = 0.07) and increase PUFA (P = 0.08) and SFA (p = 0.06).

L232; There was a tendency for interaction (p < 0.10) between annatto seed and Linseed oil on AG n-6, in which oil supplementation increased AG n-6 in the absence of annatto but in the presence of annatto seed this fatty acid was reduced (Table 5). --- There was a tendency for interaction between annatto seed and Linseed oil on AG n-6 (p < 0.10), in which oil supplementation increased AG n-6 in the absence of annatto but in the presence of annatto seed this fatty acid was reduced (Table 5).

L233; What is AG ?

L235; There was also around 176% of increase (p <0.01) on n-3 concentration in milk fat and the n-6 / n-3 ratio was improved by linseed oil supplementation, once the linseed oil reduced (p < 0.01)then by 58%. --- There was also around 176% of increase on n-3 concentration in milk fat (p < 0.01) and the n-6 / n-3 ratio was improved by linseed oil supplementation, once the linseed oil reduced (p < 0.01) then by 58%.

L241; linseed oil supplementation tended to increase (p = 0.08) HDL-cholesterol. --- linseed oil supplementation tended to increase HDL-cholesterol (p = 0.08).

Table 3. --- 
18.20, 33.60, 106.4, 2.30, 59.50
2MUN - Milk urea nitrogen; 3TS—Total solids --- OK ? 2TS, 3MUN ?

Table 4. ---
Anna3 --- Check format
Footnote --- 1 CLA ?
a-d Fisher --- a-b Fisher ?
probability.). --- probability.

Table 5. ---
a-d Fisher --- a-b Fisher ?
probability . --- probability.

Table 6. ---
HDL --- HDL ? HDL-cholesterol ?
CAT --- TAC ?
4,792 --- 4.792
Footnote --- 2CAT - 3CAT ?
linseed oil;. --- linseed oil.

L252; dry matter --- DM
L254; taste buds --- OK ?
L264; 2.5% of Linseed --- 2.5% of linseed
L271; DMI --- first appearance ?
L278; HDL --- HDL ? HDL-cholesterol ?
L281; Lima et al [47] (2016). --- Lima et al. [47].
L282; kg of DM lactating cows --- I think it is better to insert some word between DM and lactating.
L284; HDL-cholesterol --- HDL-cholesterol ? HDL ?
L304; kg MS --- What is MS ?
L321; ethereal extract --- OK ?
L322; et al [58] --- et al. [58]
L322; ether extract --- EE
L323; short-chain fatty acids (SCFA) --- SCFA
L329; C10: 0, C12: 0, C13 : 0, C14: 1, C15: 0, C16: 0) and led to a tendency to reduce C14: 0 --- C10:0, C12:0, C13:0, C14:1, C15:0, C16:0) and led to a tendency to reduce C14:0
L332; C18: 0, C18: 1 n9t, C18: 2 n6t, C18: 3 n3, C20: 3 n6, C20: 4 n6, and C21: 0 --- C18:0, C18:1 n9t, C18:2 n6t, C18:3 n3, C20:3 n6, C20:4 n6, and C21:0
L334; C18: 3 --- C18:3
L337; fat. [60]. --- fat [60].
L338; Hino et al. (1993) --- Hino et al. [   ] --- Do you cite ?
in vitro --- Italic
L343; C18: 2 --- C18:2
L351; C18: 3 --- C18:3
L357; known[2] --- known [2]
L359; Organization (2003). --- Organization [   ]. --- Do you cite ?

L402; 70, 165–178. https://doi.org/10.1111/1471-0307.12359.
L404; 84, Suppl. 1, S103-S110.
L408; 88, 1431–1441.
L421; Please confirm. I feel strange.
L444; Vyncke W. Direct determination of the thiobarbituric acid value in trichoracetic acid extracts of fish as a measure of oxidative rancidity. Fette Seifen Anstrichm. 1970, 72, 1084–1087. https://doi.org/10.1002/lipi.19700721218.
L449; Please confirm. I feel strange.
L499; Pleguezuelos, F. J., de la Fuente, L. F. and Gonzalo, C. Variation in milk yield, contents
and incomes according to somatic cell count in a large dairy goat population. J. Adv. Dairy
Res. 2015, 3, 145, doi:10.4172/2329-888x.1000145.

Author Response

Please see attachement.

Round 2

Reviewer 1 Report

Despite introduction and material and methods section have been improved as compared with the previous version, this reviewer still found the discussion hard to follow in the present form. The language and the form of the manuscript still require extensive editing in this section. The conclusion statement is very short and could be improved too.

L35 “Without influencing” or “with no effect on”

L36 “the medium chain fatty acids in milk” or “the milk medium chain fatty acids content”

L37 n-3 FA content of milk

L39 n-3 FA

L44 “The attention and consumption of foods with functional properties are increasing as” I will suggest deleting this sentence. Starting the introduction with “Food enriched…. “ will make your introduction more cohesive and to the point. If you want to keep this sentence, consider adding a period to separate it.  

L 45 Move both references at the end of the sentence.

L49 A reference is required

L 50 rich in

51 in the PUFA content of

L55 “and can predispose to metabolic diseases” unclear. Is this sentence referred to the cow supplemented with PUFAs or to the consumers of a PUFAs-enriched milk?

L55-58 Split this long sentence in two

L61 the supply…increases

L62 Unclear the reason why the authors focused their attention on vitamin C (is vitamin C contained in annatto in high quantity?). The reference you provided to support this sentence does not refer to the supplementation of vitamin C to the ruminant’s diets in order to increase vitamin C content of milk. It only mentions the addition of vitamin C to the milk or yoghurt (that is pretty different). Vitamin C is essential in human nutrition, but most of the mammals (including cattle) can synthetize it in the liver (Padilla et al., 2006). I’m not sure on the classification of vitamin C as a “carotenoid”. Please solve these points about vitamin C.

L 106-120 grounded

L 106-107 Unclear. Rephrase

L127 Blood total antioxidant capacity (here and throughout the paper)

L129-131 Providing the color of the tube top is not essential. It could be preferred to indicate the type of anticoagulant contained in the tube (i.e. heparinized tubes/ Na citrate etc.). add serum tube at the end of this sentence

L 153-166 A connection between the two sentence is missing (i.e determination of the reducing…)

L157 Use concentrations rather then levels

L173 Total antioxidant activity (here and throughout the paper)

L195 transferred into

L197 five

L212 “remained from 10 °C to 10 °C to 200 °C”unclear

Results:

My comment on moving the p value at the end of the sentences was misunderstood. I will make an example for clarity: “There was no interaction between annatto seed and linseed oil (p ≥ 0.24) on DM and nutrient intake with exception for EE intake (p = 0.03) (Table 2).” In the present form it is unclear at which comparison the first p value is referred to. Please change to: “There was no interaction between annatto seed and linseed oil  on DM and nutrient intake (p ≥ 0.24 for both), with exception for EE intake (p = 0.03) (Table 2).” Please, modify the whole results section accordingly.

L275 the dry matter intake in previous studies

Author Response

Please see attachement:

Reviewer 3 Report

I need your sincere responses.

L85; Dry matter --- dry matter
L133; HDL-C cholesterol --- HDL-cholesterol
L215; Brazil --- Brazil)

L226; My comment: It has been theoretically confirmed that the LSD method is not suitable as a multiple comparison method. I think you should use Tukey, Dunnett, Duncan for multiple comparison especially among more than 3 treatments.

Your answer: Fisher's method (LSD) is one used to compare all pairs of averages. This method controls the error rate at the level of significance α for each two-to-two comparison, but it does not control the error rate of the experiment. Fisher's procedure consists of performing multiple t tests, each at the level of significance α, only if the preliminary F test is significant at the level α.

My comment2: I feel you understand Fisher's LSD does not correct for multiple comparisons. Please respond my comment. I cannot believe scientifically all your results in Tables even now.

L275; affect the intake dry matter (DMI) ---affect the DMI
L577; 68. Organization, W.H. Diet ......Vol. 916;. --- Please confirm format.

Author Response

Please see attachement:

This manuscript is a resubmission of an earlier submission. The following is a list of the peer review reports and author responses from that submission.

Round 1

Reviewer 1 Report

This article titled "Annatto seed as antioxidants source with linseed oil for dairy cows" focuses on the effect of the combination of annatto seeds and linseed oil on milk yield, composition, antioxidant activity, nutrient digestibility, and fatty acid content of dairy cows. The experimental method of this paper was well developed, but the experimental design lacked rationality, and four Holstein cows were tested as four treatments, lacking repeated evaluations, which could easily raise doubts about the rationality of the experimental results, even though the addition of 15 g/kg dry matter of annatto seeds could reduce feed intake and milk production. In addition, the discussion part of the article is relatively superficial for the analysis of the causes of the experimental results, lacking systematic and profound discussion, and the results are consistent with the results of previous studies, lacking innovation. There are misspellings in some places, such as line 185. Overall, it is not suitable for publication in this journal.

Author Response

On the other hand, several studios have already been published using a Latin square with cows in lactation, for the present I had 16 observations, 15 degrees of freedom, where 3 degrees were for the annatto seed effect, linseed oil effect and interaction annatto seed and oil.  3 degrees for period, 3 degrees for animals and 6 degrees for residue, showing that a Latin square is a good tool for studies of digestibility and measurement of some parameters such as milk composition, oxidation products and antioxidant capacity. Currently, there are few studies on ruminants using carotenoids as an antioxidant source in the feeding of dairy cows. As described in the discussion, the cause of reduced dry matter consumption is related to terpenoids. Terpenoids and terpenes can be stored in the form of essential oils and released through structures such as secretory glands and trichomes, consequently, these compounds are concentrated in the taste buds. The main essential oil components of annatto seed are two monoterpenes called α-Pinene and β-Pinen. The effect of some volatile compounds on the consumption of alfalfa pellets by sheep, α-Pinene was depressant for the consumption of alfalfa pellets, possibly the monoterpene α-Pinene may be associated with the animal rejection of seed annatto.

Reviewer 2 Report

General Comments

This paper covers an interesting topic, with possible implication on both animals and human health. Despite that, this reviewer has several concerns about this manuscript.

The authors included testing the oxidative stability of the milk among the study objective. Despite that, analysis were performed on fresh milk, thus this reviewer cannot see the connection. Oxidative stability suggests the capacity of milk to cope with oxidation over time. What the authors seems to have tested in this paper is the antioxidant capacity of milk.

Despite performing a Latin square experimental design could increase the statistical power of the model, this reviewer has several concerns about the small experimental population used in this experiment. Are 4 cows enough to actually test the study hypothesis? Considering the individual variation observed with the biomarkers employed, this reviewer would like to see a power analysis conducted to actually know what is the power of the study with a precision of 95%.

The objective of this study require a complete rephrasing.

Please see specific comments:

SIMPLE SUMMARY:

L19 Use “increase” or “decrease” instead of “affected”

ABSTRACT:

Variables investigated, samples collected, analysis performed and statistical model adopted have not been mentioned in the abstract. It was concluded that annatto seeds did not affect the milk antioxidant capacity, but no mention to the parameters tested in this respect and neither on the results obtained was done throughout the abstract.

L31-32: define acronyms at their first use (SCFA, MCFA, LCFA)

INTRODUCTION:

L41: put “PUFA”within brackets

L45-46: rephrase this sentence

L46-47: “Indeed, oilseeds and their oils are energy sources for dairy cows, replacing starch.” Unclear the connection with previous sentence

L48 Why “making milk more susceptible to oxidation”should be desirable for human health?

L54-56: Please, notice that most of these compounds does not exert a primary antioxidant role. Their antioxidant action is mostly related to serving as co-factor for the enzymatic complexes (i.e. Selenium for GPX).

L63 Delete “it”

L65-68: This reviewer fully disagrees with the fact that linseed oil administration could induce oxidative stress since it is a PUFA-rich feed. Please, notice that oxidative processes occurring in the organisms are under a homeostatic control in normal conditions. Oxidative damages (on PUFAs as well as other molecules) occur when oxidant species exceed the antioxidant power of antioxidant molecules. This could be a consequence of a lack of antioxidants (i.e. insufficient exogenous antioxidants supply or ineffective endogenous antioxidants functioning) or an excessive production of oxidant species. Among others, a cause of increased oxidant species production is immune cells functioning and inflammatory process. Since linseed oil is a n-3 PUFA reach feed (and considering antinflammatory role exerted by n-3 PUFAs on immune cells), then a lower production of oxidant species could be expected when linseed oil is administered to cows.

L69: “in diets”and “in dairy cows diets”are erpetitions

L68-71: Unclear the connection of “dry matter intake (DMI), digestibility of dry matter and nutrients, milk yield, composition, and fatty acid profile” with the oxidative damages on animal organism and milk.

MATERIALS AND METHODS:

As a style comment, the readability of your methods section could benefit from splitting it into subsections reporting both sample collection and analysis performed for each variable considered (i.e.  experimental design and animal management, blood collection and analysis, milk samples collection and analysis etc.)

Further detail on the experimental design adopted in this trial are required. How long did each phase last? Did you perform any washout period between phases? How long did the whole experiment last?

Didn’t you perform any baseline for measured parameters?

How was dry matter intake calculated? How was nutrient digestibility calculated?

L76: further details on animals enrolled are required: body weight, BCS, parity (number of lactations), average milk yield (kg/d), milk yield in the previous lactation. Please, include these detail as mean +/- SD

L78: close brackets after annatto seeeds

L78: are 600 g/kg on a dry matter basis?

L81: correct “annatto”

L98: Why you choose different timing for feces collection? Please provide a reference for such a collecting protocol.

L102: Why wasn’t blood collected in fasting conditions?

L108: “For statistical analysis, only the data referring to the collection periods were used.” Unclear

L112: why you used a preservative? How long was the milk stored before analysis?

L122 NDF

L125 Define HDL and TAC at first use

L 136: please, define modifications.

L173: what you mean by 1 and 2? With and without?

L 179 at

Statistical analysis: Unclear to me what you mean by “period”and how long each period exactly lasted. Did you run a MIXED using 2 random effects and no repeated measures? What about time? You collected multiple samples for milk, feces etc. How did you accounted for repeated measurements?

RESULTS

Throughout results, move P at the end of the sentences (i.e. “…for EE intake (P = 0.03)” instead of “…(P = 0.003) for EE intake”).

L 185 In; affect (repetition)

Table 3: Is milk composition reported on a dry matter basis?

L202: a period is missed

L206: Delete bracket after MUFA

L207: Define AG

Table 2: Correct NDF

Author Response

General Comments

This paper covers an interesting topic, with possible implication on both animals and human health. Despite that, this reviewer has several concerns about this manuscript.

The authors included testing the oxidative stability of the milk among the study objective. Despite that, analysis were performed on fresh milk, thus this reviewer cannot see the connection. Oxidative stability suggests the capacity of milk to cope with oxidation over time. What the authors seems to have tested in this paper is the antioxidant capacity of milk.

Despite performing a Latin square experimental design could increase the statistical power of the model, this reviewer has several concerns about the small experimental population used in this experiment. Are 4 cows enough to test the study hypothesis? Considering the individual variation observed with the biomarkers employed, this reviewer would like to see a power analysis conducted to know what the power of the study with a precision is of 95%.

Answer:

I agree with the reviewer that the oxidative stability term is not suitable for this article, in the study we determined oxidation products and antioxidant activity (ABTS), for that reason change the term in table 3 for oxidative products and antioxidant activity. Oxidative stability is used to evaluate milk for a certain period. On the other hand, several studios have already been published using a Latin square with cows in lactation, for the present I had 16 observations, 15 degrees of freedom, where 3 degrees were for the annatto seed effect, linseed oil effect and interaction annatto seed and oil.  3 degrees for period, 3 degrees for animals and 6 degrees for residue, showing that a Latin square is a good tool for studies of digestibility and measurement of some parameters such as milk composition, oxidation products and antioxidant capacity.

The objective of this study requires a complete rephrasing.

Answer:

Objectives of this study were to evaluate the effects of dietary of annatto seed, linseed oil, and their combination on dry matter intake and nutrient apparent total tract digestibility, as well as changes in the antioxidant capacity, chemical, and fatty acid composition of milk in lactating cows.

Please see specific comments:

SIMPLE SUMMARY:

L19 Use “increase” or “decrease” instead of “affected”

Answer: this contribution has been met. I thank the reviewer for his contribution.

ABSTRACT:

Variables investigated, samples collected, analysis performed and statistical model adopted have not been mentioned in the abstract. It was concluded that annatto seeds did not affect the milk antioxidant capacity, but no mention to the parameters tested in this respect and neither on the results obtained was done throughout the abstract.

Answer:

The objective makes very clear the variables to be studied and analysis “The objectives of this study were to evaluate the effects of dietary of annatto seed, linseed oil, and their combination on dry matter intake and nutrient apparent total tract digestibility, as well as changes in the antioxidant capacity, chemical, and fatty acid composition of milk in lactating cows”. The statistical model is described in the summary, highlighting the most important information of the results so that the article becomes more attractive. Within the results I put that the annatto seed does not affect the antioxidant activity in milk

L31-32: define acronyms at their first use (SCFA, MCFA, LCFA)

Answer: this contribution has been met. I thank the reviewer for his contribution.

INTRODUCTION:

L41: put “PUFA”within brackets

Answer: this contribution has been met. I thank the reviewer for his contribution.

L45-46: rephrase this sentence

Answer:

Dairy cows’ diets supplemented with linseed oil rich source of C18: 3n-3 (α-linolenic acid) has been shown to increase milk PUFA [5]

L46-47: “Indeed, oilseeds and their oils are energy sources for dairy cows, replacing starch.” Unclear the connection with previous sentence.

Answer:  The sentence has been removed from the paragraph.

L48 Why “making milk more susceptible to oxidation”should be desirable for human health?

Answer:

In contrast, the nutritional improvement of milk with an increase in PUFA content makes it more susceptible to oxidation. Oxidative damage is caused by the chain reaction of free radicals in fat membranes. PUFAs have double bonds that are susceptible to electron losses, triggering lipoperoxidation of PUFAs.

L54-56: Please, notice that most of these compounds does not exert a primary antioxidant role. Their antioxidant action is mostly related to serving as co-factor for the enzymatic complexes (i.e. Selenium for GPX).

Answer:

Clearly that they participate as co-factors, but the phrase in the context is that through diets we can add foods, supplements and additives with antioxidant potential or as co-factors.

L63 Delete “it”

Answer: this contribution has been met. I thank the reviewer for his contribution.

L65-68: This reviewer fully disagrees with the fact that linseed oil administration could induce oxidative stress since it is a PUFA-rich feed. Please, notice that oxidative processes occurring in the organisms are under a homeostatic control in normal conditions. Oxidative damages (on PUFAs as well as other molecules) occur when oxidant species exceed the antioxidant power of antioxidant molecules. This could be a consequence of a lack of antioxidants (i.e. insufficient exogenous antioxidants supply or ineffective endogenous antioxidants functioning) or an excessive production of oxidant species. Among others, a cause of increased oxidant species production is immune cells functioning and inflammatory process. Since linseed oil is a n-3 PUFA reach feed (and considering antinflammatory role exerted by n-3 PUFAs on immune cells), then a lower production of oxidant species could be expected when linseed oil is administered to cows.

Answer:

The supplementation of PUFAs is very desirable for the body and improves the fatty acid profile in milk and is an adjunct to reducing inflammatory processes, omega-3 and omega-6 fatty acids inhibit the production of inflammatory prostaglandins and stimulate the production of prostaglandins series 1 and 3 (PGE1 and PGE3). However, this fat profile is unstable and susceptible to the occurrence of rancid taste in milk. Table (1) shows an increase in the ether extract in linseed oil treatments compared to control treatment, leading to a predisposition to undergo lipoperoxidation and exposing animals to deleterious effects of oxidative stress.

L69: “in diets”and “in dairy cows diets”are erpetitions

Answer: this contribution has been met. I thank the reviewer for his contribution.

L68-71: Unclear the connection of “dry matter intake (DMI), digestibility of dry matter and nutrients, milk yield, composition, and fatty acid profile” with the oxidative damages on animal organism and milk.

Answer:

Therefore, the objectives of this study were to evaluate the effects of dietary of annatto seed, linseed oil, and their combination on dry matter intake and nutrient apparent total tract digestibility, as well as changes in the antioxidant capacity, chemical, and fatty acid composition of milk in lactating cows. We hypothesized that linseed oil supplementation would increase n-3 concentration while annatto seed would increase antioxidant capacity and consequently decrease milk oxidation, and the association between the two treatments would increase n-3 concentration in the milk in conjunction with lower oxidation.

MATERIALS AND METHODS:

As a style comment, the readability of your methods section could benefit from splitting it into subsections reporting both sample collection and analysis performed for each variable considered (i.e.  experimental design and animal management, blood collection and analysis, milk samples collection and analysis etc.)

Further detail on the experimental design adopted in this trial are required. How long did each phase last? Did you perform any washout period between phases? How long did the whole experiment last?

Answer:

Used in a 4 × 4 Latin square design with a 2 × 2 factorial arrangement (with or without annatto seed; with or without linseed oil), with four experimental periods of 21 days, with 16 days for adaptation and 5 days for collection, totaling 84 experiment days.

Didn’t you perform any baseline for measured parameters?

Answer:

Baseline for the measured parameters was used for blood chemistry samples.

How was dry matter intake calculated? How was nutrient digestibility calculated?

Answer:

The consumption of dry matter is calculated according to the dry matter of the consumed * kg - the dry matter of the offered * kg. Regarding the digestibility calculation, it was estimated according to the iNDF of the diets, leftovers and feces through incubation in situ for 240 hours. Afterwards, the production of fecal dry matter was estimated, which allowed the calculation of fecal excretion of each nutrient, the excretion value will be deducted from the consumption of each nutrient divided into the consumption of each nutrient multiplied 100

L76: further details on animals enrolled are required: body weight, BCS, parity (number of lactations), average milk yield (kg/d), milk yield in the previous lactation. Please, include these detail as mean +/- SD

Answer:

The experiment was conducted at the State University of Maringa (UEM), Brazil. The experimental protocol was approved by the UEM Ethics Committee (number 6450240117). Four multiparous Holstein (120 ± 43 days in milk, 15.98 ± 2.02 of milk/day, 566 ± 64 kg of body weight, mean ± SD).

L78: close brackets after annatto seeeds

Answer: this contribution has been met. I thank the reviewer for his contribution.

L78: are 600 g/kg on a dry matter basis?

The control diet was composed of 600 g/kg (DM basis) corn silage, and 400 g/kg (DM basis) concentrate, and was composed of soybean meal, ground corn grain, mineral and vitamin, molasses powder and Limestone.

L81: correct “annatto”

Answer: Correct

L98: Why you choose different timing for feces collection? Please provide a reference for such a collecting protocol.

Answer:

Feces samples were taken every 9 hours for 72 hours, to have a more representative sample during different hours of the day.

C.B. Sampaio, E. Detmann, T.N.P. Valente, V.A.C. Costa, S.C. Valadares Filho, A.C. Queiroz Fecal excretion patterns and short term bias of internal and external markers in a digestion assay with cattle Rev. Bras. Zootec., 40 (2011), pp. 657-665

L102: Why wasn’t blood collected in fasting conditions?

Answer:

Our research group worked with different antioxidants erva mate, propolis, flaxseed flour and citrus pulp, we show that the best peaks of antioxidants are between 3 to 4 hours after feeding.

L108: “For statistical analysis, only the data referring to the collection periods were used.” Unclear

Answer:

In each period, there was an adaptation period of 16 days, 5 days for collection of stool samples, rejections and diets, milk and blood. The values of the analyzes during each collection period were used for statistical analysis.

L112: why you used a preservative? How long was the milk stored before analysis?

Answer:

The second aliquot, around 100 mL of milk, without the addition of preservative, was frozen at -80 °C for the subsequent determination of the concentration of fatty acids and antioxidant parameters.  Analysis was performed 20 days after finishing the experiment for antioxidants and fatty acid profile.

L122 NDF

Answer: this contribution has been met. I thank the reviewer for his contribution.

L125 Define HDL and TAC at first use

Answer: this contribution has been met. I thank the reviewer for his contribution.

L 136: please, define modifications.

Answer: this contribution has been met. I thank the reviewer for his contribution.

L173: what you mean by 1 and 2? With and without?

Answer: this contribution has been met. I thank the reviewer for his contribution.

L 179 at

Answer: this contribution has been met. I thank the reviewer for his contribution.

Statistical analysis: Unclear to me what you mean by “period”and how long each period exactly lasted. Did you run a MIXED using 2 random effects and no repeated measures? What about time? You collected multiple samples for milk, feces etc. How did you account for repeated measurements?

Answer:

In the Latin Squares design, local control is adopted in two directions. Each treatment appears only once in each row as a fixed effect, and in each column the period appears only once as a random effect. Furthermore, another random effect is the animal, so with the fact that only having a random effect can use the mixed model, it is not a requirement to have repeated measures over time to use this type of statistical tool.

RESULTS

Throughout results, move P at the end of the sentences (i.e. “…for EE intake (P = 0.03)” instead of “…(P = 0.003) for EE intake”).

Answer: this contribution has been met. I thank the reviewer for his contribution.

L 185 In; affect (repetition)

Answer: this contribution has been met. I thank the reviewer for his contribution.

Table 3: Is milk composition reported on a dry matter basis?

Answer:

It was a mistake in writing.

L202: a period is missed

Answer: this contribution has been met. I thank the reviewer for his contribution.

L206: Delete bracket after MUFA

Answer: this contribution has been met. I thank the reviewer for his contribution.

L207: Define AG

Answer: this contribution has been met. I thank the reviewer for his contribution.

Table 2: Correct NDF

Answer: this contribution has been met. I thank the reviewer for his contribution.

Reviewer 3 Report

The objective and design is innovative. However the value of the research is compromised by only four cows in a single Latin square. The statistical analysis requires further description. It appears that only one mean value is used for each cow in each period giving only 15 DF. Were repeated measures for dome measurements considered.

The cow size and milk production level at early lactation require discussion as both are much lower than expected for the Holstein breed. Was feed restricted as in the paper by Lima et al. (Acta Scientiarum, Animal Science 42:e47651, 2020). Is annatto seed palatability an issue. Based on this reviewer’s application of the Cornell Model 6.5 in the NDS format and the CNCPS feed data base the control ration met energy and protein requirements and NDF intake was not limiting dry matter intake. Discussion of a limit to DM intake would be useful. Include treatment body weight change if available. Also body condition score.

Determination of digestibility was useful, however, examination of the digestibility of annatto seeds or bixin would have been useful in explaining annatto effects. As the annatto seed is small and with a relatively hard coat at least 30% may have escaped rupture during mastication and rumination. This 30% estimate is based on cereal grain fecal levels. Was the seed processed making this comment irrelevant. Please discuss in relation to oxidative/antioxidant profile in Table 3. The authors should note the possible extensive intestinal metabolism of carotenoids as this may affect post absorptive effects. See for example Japan Agric Res Quarterly 48(4):385, 2014.

Table 2. Is FDN meant to be NDF. Was NDF measured as aNDF or aNDFom.

Line 236. Apparent digestibility, a diet fat level and metabolic fecal fat are issues in true digestibility.

Line 265. Clarify, Percent, not g.

Line 282 and Table3. Trend (not significant) at P=0.07. DC or CD.

Line 326-27. Reword.

Line 335. Reduced.

Line 335 and 377. 30g/kg of ration DM.

Comprehensive references are appreciated.

Author Response

  1. The objective and design is innovative. However the value of the research is compromised by only four cows in a single Latin square. The statistical analysis requires further description. It appears that only one mean value is used for each cow in each period giving only 15 DF. Were repeated measures for dome measurements considered.

Answer:

Several studios have already been published using a Latin square with cows in lactation, for the present I had 16 observations, 15 degrees of freedom, where three degrees were for the annatto seed effect, linseed oil effect and interaction  annatto seed and oil, 3 degrees for period, 3 degrees for animals and 6 degrees for waste, showing that a Latin square is a good tool for studies of digestibility and measurement of some parameters such as milk composition, oxidation products and antioxidant capacity. Repeated measures over time on the same animal were not considered.

  1. The cow size and milk production level at early lactation require discussion as both are much lower than expected for the Holstein breed.

Answer:

The number of animals was 4, milk production at the time was low for the Holstein breed, but at that time it was animals that are available for research.

  1. Was feed restricted as in the paper by Lima et al. (Acta Scientiarum, Animal Science 42:e47651, 2020). Is annatto seed palatability an issue. Based on this reviewer’s application of the Cornell Model 6.5 in the NDS format and the CNCPS feed data base the control ration met energy and protein requirements and NDF intake was not limiting dry matter intake. Discussion of a limit to DM intake would be useful. Include treatment body weight change if available. Also body condition score.

Answer:

The weights of the animals had no difference, so the discussion was focused on the terponoids.  Terpenoids and terpenes can be stored in the form of essential oils and released through structures such as secretory glands and trichomes, consequently, these compounds are concentrated in the taste buds. The main essential oil components of annatto seed are two monoterpenes called α-Pinene and β-Pinen. The effect of some volatile compounds on the consumption of alfalfa pellets by sheep, α-Pinene was depressant for the consumption of alfalfa pellets, possibly the monoterpene α-Pinene may be associated with the animal rejection of seed annatto

  1. Determination of digestibility was useful, however, examination of the digestibility of annatto seeds or bixin would have been useful in explaining annatto effects. As the annatto seed is small and with a relatively hard coat at least 30% may have escaped rupture during mastication and rumination. This 30% estimate is based on cereal grain fecal levels. Was the seed processed making this comment irrelevant. Please discuss in relation to oxidative/antioxidant profile in Table 3.

Answer:

The annatto seed was not processed, as it is a seed rich in carotenoids, I avoided exposing the material to light and processes that can potentiate the lost of antioxidants. Grinding a sample would be ideal to improve the digestibility of the annatto seed, but the intention was that the bixin is in the pericarp that is outside the seed. The addition of annatto seed did not affect the oxidative products and antioxidant activity, as the seed was included with the objective of transferring its polyphenolic compounds to milk, in order to decrease oxidation of milk fat that was enriched with PUFA. Regarding the antioxidant activity in blood and milk, lipid supplementation increased CD concentration due to increased PUFA intake, which double bonds that predispose to lipoperoxidation the cause of electron loss, thus increased CD in milk.

  1. The authors should note the possible extensive intestinal metabolism of carotenoids as this may affect post absorptive effects. See for example Japan Agric Res Quarterly 48(4):385, 2014.

Answer:

Absorption and transport processes of carotenoids are similar to those of lipids. After ingested, the carotenoids are incorporated in mixed micelles consisting of bile acids, free fatty acids, monoglycerides and phospholipids. However, there are nutrients that, when ingested with carotenoids, interfere with the absorption and metabolism process, in this study we use flaxseed oil that would help the absorption of carotenoids according to a study by Dimitrov et al. (1998) in a study in humans found that absorção-carotene absorption is affected by the concentration of dietary fat in receiving a high-fat diet, compared to individuals who received a low-fat diet.

  1. Table 2. Is FDN meant to be NDF. Was NDF measured as aNDF or aNDFom.

Answer:

There was an error in the writing, correct acronym is NDF, only if I measure NDF and iNDF

  1. Line 236. Apparent digestibility, a diet fat level and metabolic fecal fat are issues in true digestibility.

Answer:

Fat and protein can influence true digestibility, especially because it has a metabolic loss in the stool, compared to fiber and carbohydrates

Line 265. Clarify, Percent, not g.

  1. Line 282 and Table3. Trend (not significant) at P=0.07. DC or CD.

CD

  1. Line 326-27. Reword.

Answer: this contribution has been met. I thank the reviewer for his contribution.

  1. Line 335. Reduced.

Answer: this contribution has been met. I thank the reviewer for his contribution.

  1. Line 335 and 377. 30g/kg of ration DM.

Answer: this contribution has been met. I thank the reviewer for his contribution.

  1. Comprehensive references are appreciated.
